# Constant Rate Infusion of Lidocaine, Tumescent Anesthesia and Their Combination in Dogs Undergoing Unilateral Mastectomy

**DOI:** 10.3390/ani11051280

**Published:** 2021-04-29

**Authors:** Cecilia Vullo, Adolfo Maria Tambella, Annastella Falcone, Gabriele Marino, Giuseppe Catone

**Affiliations:** 1Department of ChiBioFarm, University of Messina, 98168 Messina, Italy; 2School of Bioscience and Veterinary Medicine, University of Camerino, 62024 Matelica, Italy; adolfomaria.tambella@unicam.it; 3Department of Veterinary Sciences, University of Messina, 98168 Messina, Italy; annastella.falcone@unime.it (A.F.); gabriele.marino@unime.it (G.M.); gcatone@unime.it (G.C.)

**Keywords:** analgesia, canine, local anesthetic, mammary gland, tumor

## Abstract

**Simple Summary:**

Appropriate pain management, particularly when large tissue is removed, allows for better recovery during the postoperative period. The analgesic effects of regional tumescent anesthesia (TUM) combined with constant rate infusion (CRI) of lidocaine were evaluated in dogs submitted to unilateral mastectomy. The purpose of this study was to evaluate if the addition of TUM to lidocaine CRI influenced cardiopulmonary function in dogs undergoing unilateral mastectomy and provided adequate postoperative analgesia. The authors concluded that the combination of the two technique decreases pain and requirement for rescue analgesia than with lidocaine CRI or TUM alone.

**Abstract:**

Tumescent anesthesia (TUM) is a technique that was initially used to perform liposuction under local anesthesia, which consists of the injection of such large volumes of local anesthetic until to produce swelling and firmness (tumescence) of the surgical area. The aim of this study was to compare the intraoperative analgesic efficacy of lidocaine (LID) constant rate infusion (CRI), of TUM, or their combination (LID/TUM) and the postoperative pain and analgesic requirement in dogs undergoing unilateral mastectomy. Twenty-four dogs were premedicated with dexmedetomidine (3 μg/kg) and methadone (0.2 mg/kg) intravenously (IV). After induction with propofol IV to effect, dogs were randomly allocated to receive a loading dose of lidocaine (2 mg/kg) followed by a CRI of 100 μg/kg/min (Group LID) in addition to an equivalent volume of lactated Ringer’s solution instead of local TUM; a loading dose of lactated Ringer’s solution followed by a CRI of Ringer’s solution in addition to TUM (Group TUM); a loading dose of lidocaine (2 mg/kg) followed by a CRI of 100 μg/kg/min in addition to TUM (Group LID/TUM). Anesthesia was maintained with isoflurane in oxygen. Postoperative pain scores were assessed once the dogs had fully recovered from the sedative effects, and following 15, 30, 45 and 60 min. The results of the current study allow us to assert that all three treatments provided satisfactory intraoperative antinociceptive effects but administration of LID/TUM induced greater inhibition on sympathetic stimulating effect up to 60 min from recovery, thus, providing better early postoperative pain relief in dogs undergoing mastectomy.

## 1. Introduction

The mammary gland is a modified apocrine sweat gland, and it is the most common site for the development of benign and malignant tumors in intact female dogs [1]. Sites of metastases are regional lymph nodes and lungs, although other organs may also be involved [2]. Mastectomy remains the gold-standard treatment for most types of these tumors excluding inoperable highly metastatic disease and most of the inflammatory mammary carcinomas [1,3]. Postoperative pain followed the surgery may consist of inflammatory, neurogenic, and visceral components [4]. To minimize perioperative patients’ distress, especially after major surgery, as mastectomy, requires multimodal pre-emptive analgesia that includes both systematical and local or regional administration of analgesics [5,6].

Lidocaine is an amide local anesthetic and antiarrhythmic agent [7]. The intravenous (IV) use or constant rate infusion (CRI) of lidocaine as a supplement to general anesthesia has been reported in dogs [8,9,10,11,12,13,14,15,16,17,18], and has been used for surgical procedures, as delineated in recent veterinary pain management guidelines because it plays an important role in the control of peri and postoperative sympathetic response [19,20].

Tumescent anesthesia (TUM) is a technique for regional anesthesia of the skin and the subcutaneous tissues, using direct infiltration of large volumes of diluted local anesthetic combined with a vasoconstrictor, described for the first time in 1987 [21,22]. The infiltration of the tumescent local anesthetic solution may be performed by Klein’s cannula connected to a syringe alone or an infusion pump [23]. This technique of injection was described in bitches and in cats that underwent a unilateral mastectomy, demonstrating the effectiveness of the technique to facilitate the surgical procedure and to assure satisfactory postoperative analgesia [24,25,26].

The purpose of this study was to evaluate if the addition of TUM to lidocaine CRI modified intraoperative cardiopulmonary function in dogs undergoing unilateral mastectomy and provided adequate early postoperative analgesia. We hypothesized that the combination of lidocaine CRI with TUM would decrease pain and requirement for rescue analgesia than with lidocaine CRI or TUM alone.

## 2. Materials and Methods

The study was approved by the Bioethics Committee of the Department of Veterinary Sciences of the University of Messina following Good Scientific Practice guidelines and national legislation (Approval N. 047/2021). Informed consent was obtained from the owner of the dogs included in this study.

The sample size was calculated using the Analysis of Variance method with power 80%, alpha-error of 0.05 and effect size (f = 0.69) obtained from unpublished preliminary data. The power analysis was performed with G-Power software, version 3.1.9.2.

Twenty-four mixed-breed neutered female dogs aged six to fourteen years and with bodyweight between 7 and 22 kg presented to the Veterinary Teaching Hospital, University of Messina, for unilateral mastectomy due to mammary tumors were included in a randomized, prospective, blinded clinical study. The preoperative condition of each dog was evaluated through thoracic radiographs, a physical examination (i.e., behavior, mucous membranes, hydration status, temperature, cardiopulmonary auscultation, heart rate, respiratory rate, capillary refill time), and laboratory tests. Dogs were excluded from the study if they were not spayed, were obese or had abnormal laboratory data, arterial hypertension, congestive heart failure, renal or hepatic dysfunction, pulmonary metastases, inflamed or ulcerated tumors, infiltrating large masses (over 5 cm), or an ASA health status of greater than III.

Food, but not water, was withheld for at least 10 h before anesthesia. A 20 or 22-gauge catheter was aseptically placed in a cephalic vein and all dogs were premedicated with a neuroleptanalgesic combination of dexmedetomidine (3 μg/kg, Dexdomitor, Vetoquinol, Italy) and methadone (0.2 mg/kg, Semfortan, Dechra, Torino, Italy) mixed in the same syringe and administered intravenously. Immediately after sedation, dogs were placed on top of an electrical heating pad and irradiated with a heating lamp, until they were taken to the operating room. A second IV catheter was placed for the administration of lidocaine or placebo (Lactated Ringer’s solution, S.A.L.F., Bergamo, Italy) during general anesthesia. Induction of anesthesia was produced by administration of propofol (1.6–2 mg/kg IV, Proposure, Merial, Milano, Italy) as required to enable endotracheal intubation. The animals were connected to a rebreathing or non-rebreathing circuit according to the weight of the animal. Isoflurane in 100% oxygen was delivered for maintenance of anesthesia in spontaneous respiration. Lactated Ringer’s solution (Lactated Ringer’s solution, S.A.L.F., Italy) was administered at 10/mL/kg hour for the duration of the anesthesia.

Dogs were then placed in lateral recumbency and a 20 or 22-gauge catheter was aseptically introduced into the dorsal pedal artery for direct blood pressure monitoring and the collection of arterial blood to determine blood gases. After induction of anesthesia, the dogs were randomly assigned to one of the three following groups with 8 animals in each, using the random number generator GraphPad QuickCalcs Software (GraphPad Software Inc., San Diego, CA, USA): Group LID (*n* = 8): an IV loading dose of lidocaine (2 mg/kg, Lidocaine 2%, ATI, Bologna, Italy) followed by a CRI of 100 μg/kg/min; Group TUM (*n* = 8): an IV loading dose of lactated Ringer’s solution followed by a CRI of Ringer’s solution in addition to local TUM applied immediately before mastectomy; Group LID/TUM (*n* = 8): an IV loading dose of lidocaine (2 mg/kg followed by a CRI of 100 μg/kg/min) in addition to local TUM. Group LID received an equivalent volume of lactated Ringer’s solution instead of local TUM. All CRIs began immediately after the loading dose and were infused during the time of anesthesia using a syringe pump (Alaris^®^ GH Syringe Pump). Dogs were placed in dorsal recumbency on an electrical heating pad throughout anesthesia.

The local anesthetic solution for TUM was prepared by mixing 40 mL of 2% lidocaine plus 20 μg/mL adrenaline (Lidocaine 2%, ATI, Bologna, Italy) into a refrigerated (8 °C) lactated Ringer’s solution (250 mL, Lactated Ringer’s solution, S.A.L.F., Italy). The final local anesthetic solution contained 800 mg of lidocaine and 800 μg of adrenaline and it was administered at 12 mL/kg (240 mg/kg of lidocaine and 240 μg/kg of adrenaline) for the entire length of the mammary glands, from the thoracic to the inguinal region using a ten holes cannula (1.65 mm × 150 mm/16G × 6′′, Aesthetic Group-Z.A. La Gobette). The cannula was inserted under the skin with two incisions created cranial and caudal to the thoracic and inguinal portion of the mammary glands. Group LID received the placebo solution using the same procedure. The same qualified surgeon performed the mastectomy, using a new generation cordless ultrasonic device, the Sonicision^®^ (Medtronic, Milano, Italy). The infusions were stopped at the end of anesthesia.

The animals were connected to a multiparametric anesthetic monitor (BeneView T8, Mindray Bio-Medical Electronics Co., Ltd, Milano, Italy) and the electrocardiogram (ECG), invasive systolic (SAP), diastolic (DAP) and mean arterial blood pressures (MAP), heart rate (HR/min), respiratory rate (RR/min), arterial oxygen saturation (SpO_2_), esophageal body temperature (T °C) and end-tidal partial pressure of CO_2_ (EtCO_2_ mmHg) were continuously recorded. Arterial blood pH, arterial oxygen (PaO_2_) and carbon dioxide (PaCO_2_) tensions and bicarbonate concentration (HCO_3_^−^) were recorded immediately after the introduction of the arterial catheter (T0), immediately after the start of the surgery (T1), and at 15 (T2), 30 (T3), 40 (T4) minutes following the start of the surgery, using i-STAT System (Abbott). Capillary refill time and the peripheral pulse palpation were also continuously monitored.

Anesthetic, surgery, endotracheal extubation, and recovery times were recorded.

Subjective postoperative pain scores were attributed by a blinded evaluator, who was unaware of analgesics administered, using the Italian version of the Glasgow Composite Pain Scale-Short Form (ICMPS-SF), that included 6 behavioral categories with associated descriptive expressions (vocalization, four descriptions; attention to wound, five descriptions; mobility, five descriptions; response to touch, six descriptions; demeanor, five descriptions; posture/activity, five descriptions) [27]. The scale was applied once the dogs had fully recovered consciousness and were able of standing (RT0), and following 15 (RT1), 30 (RT2), 45 (RT3), and 60 min (RT4). In the same time frame, considering postoperative pain scores exceeding level 6/24 as clinical decision-point for the requirement of rescue analgesia, IV administration of 0.2 mg/kg methadone was provided.

Cardinal data were assessed for normality using the Shapiro-Wilk test. All cardinal variables were compared between the three groups using the One-way Analysis of Variance (ANOVA) with subsequent Holm–Sidak post-hoc test. The repeated measures ANOVA and the Holm–Sidak post-hoc tests were used to compare the study time-points within each group. The ICMPS-SF scores were analyzed between the three groups using the Kruskal–Wallis test followed by the Dunn’s multiple comparison test. The Friedman test and the Dunn’s multiple comparison test were used to compare the scores in the study time-points within each group. The Chi-square (χ^2^) and Fisher’s exact tests were used for frequency analysis (dogs requiring postoperative rescue analgesia).

Differences with *p*-values < 0.05 were considered statistically significant. All data were analyzed using the software GraphPad Prism 8 for MacOS, version 8.2.1 (GraphPad Software Inc., San Diego, CA, USA).

## 3. Results

Twenty-four mixed-breed neutered dogs collected in three months (December 2020–February 2021) met the inclusion criteria and completed the study. A three-arm randomized clinical trial with a balanced allocation ratio per group (8:8:8) was conducted.

Anesthesia and surgery were performed without complications in all cases.

The total anesthesia time ranged from 46 to 77 min, and the duration of the surgery was 38 to 66 min. There was no significant difference between the groups for age, weight, duration of surgery, anesthesia, time to endotracheal extubation, and recovery time (Table 1 and Figure 1).

During anesthesia, the measurements of invasive SAP and DAP were not significantly different between groups. On the contrary, the MAP was significantly lower in the Group LID/TUM (69.13 ± 8.228 mmHg; F = 21.35; *p* < 0.0001) in comparison to both the Group LID (82.56 ± 5.305 mmHg; *p* < 0.0001) and the Group TUM (81.87 ± 5.643 mmHg; *p* < 0.0001) (Figure 2).

There no were significant differences observed for HR, RR, SpO_2_ and EtCO_2_ between groups (Figure 3).

Oesophageal body temperature during anesthesia was statistically different between groups (F = 5.940; *p* = 0.005), with Group LID showing a higher mean temperature (36.57 ± 0.5621 °C) than Group TUM (35.99 ± 0.7232 °C; *p* = 0.0438) and Group LID/TUM (35.75 ± 0.8224 °C; *p* = 0.0047) (Figure 4).

Arterial blood pH, PaO_2_, PaCO_2_ and HCO_3_^−^ were not significantly different between groups.

In addition, there were no significant differences comparing the study time-points within each group (Figure 5).

Some differences between groups were found in postoperative pain intensity evaluated by ICMPS-SF at RT0 (H = 15.18; *p* = 0.0005), RT1 (H = 14.47; *p* = 0.0007), RT2 (H = 13.51; *p* = 0.0012), RT3 (H = 7.926; *p* = 0.0190), while at RT4 the intergroup differences faded (H = 2.426; *p* = 0.2974).

Once full recovery from the sedative effects of the anesthetic drugs was achieved (RT0), the dogs in the Group LID/TUM (mean score ± standard deviation 2.375 ± 1.923, median 2.000) showed a significantly lower pain score than Group TUM (mean score ± s.d. 6.375 ± 2.134, median 6.000, *p* = 0.0419) and a significantly lower pain score than the Group LID (mean score ± s.d. 9.125 ± 3.314, median 8.500, *p* = 0.0004).

The intergroup significant difference between LID and LID/TUM groups persisted in subsequent evaluations, after 15 min (RT1, *p* = 0.0008), 30 min (RT2, *p* = 0.0009), and 45 min (RT3, *p* = 0.0304). The intergroup significant difference between TUM and LID/TUM groups was showed up until RT1 (*p* = 0.0171). A significant difference between LID and TUM groups was only found at RT2 (*p* = 0.0464).

A tendency towards a progressive decrease of the postoperative pain score at different time points was manifested by all groups. Significant decrease in ICMPS-SF pain score were showed in group LID (χ^2^_r_ = 21.95, *p* = 0.0002) and in Group TUM (χ^2^_r_ = 28.84, *p* < 0.0001), particularly from RT0 to RT3 in TUM (*p* = 0.0050) and from RT0 to RT4 in LID (*p* = 0.0006) and TUM (*p* = 0.0006). The LID/TUM group, having lower basal postoperative scores than the other groups, showed a non-significant decrease in pain score (Figure 6).

Considering separately the item response to touch included in ICMPS-SF, no differences between groups were found. A progressive decrease tendency appeared within each group, with the lower mean score ± s.d. achieved at RT4 in all groups (LID: 0.75 ± 0.89, TUM: 0.25 ± 0.46, LID/TUM: 0.12 ± 0.35); the difference was not significant within Group LID (χ^2^_r_ = 3.958, *p* = 0.4117) while it was significant within Group TUM (χ^2^_r_ = 17.25, *p* = 0.0017) and Group LID/TUM (χ^2^_r_ = 12.00, *p* = 0.0174).

Considering the frequencies of the requirement for rescue analgesia at the postoperative time frame, a noticeable overall difference between groups was found (χ^2^ = 16.45, *p* = 0.0003). In the Group LID, all dogs required rescue analgesia (five at RT0, two at RT2 and one at RT3). In the Group TUM, five dogs required rescue analgesia (three at RT0 and two at RT1). The difference in frequencies between LID and TUM groups was not significant (*p* = 0.2000). No dog in the Group LID/TUM reached the pain score threshold for rescue analgesia, hence showing a significant difference compared to both Group LID (*p* = 0.0002) and Group TUM (*p* = 0.0256).

## 4. Discussion

Poorly controlled acute pain remains one of the most undesirable consequences after surgery. Pain control is essential for postoperative management for the surgical patient, not only for ethical reasons, but also because failure to recognize pain can lead to a number of consequences such as an increase in the incidence of complications and changes in the nervous system’s plasticity [28,29]. Pain recognition and assessment in animals are challenging because of their inability to communicate and the complexity of pain perception and variation in behavioral reactions [30]. The goal of pain treatment is blocking the generation, transmission, perception and sensation of nociceptive stimuli in different levels of the peripheral and central nervous system. Opioid analgesics are commonly used in clinical practice for postoperative pain treatment. However, its use is related to many side effects, such as respiratory depression, nausea, vomit, urinary retention and constipation [31,32]. Therefore, alternative techniques and medications have been used as a replacement for opioids for analgesia [33].

Lidocaine is an amide-type local anesthetic that produces pharmacological action by blocking the sodium channels in neural tissues, thus, interrupting neuronal transmission. Lidocaine is widely-available and commonly used as a local anesthetic. IV administration of lidocaine at doses between 50 and 200 μg/kg/min demonstrates anti-hyperalgesic properties that reduce the intraoperative and postoperative pain in dogs without causing clinically significant hemodynamic instability [15,16]. Gutierrez-Blanco et al. reported that intraoperative 100 μg/kg/min lidocaine CRI followed by another 4 h of infusion at a dose of 25 μg/kg/min resulted in inadequate postoperative analgesia after ovariohysterectomy in dogs [34]. In this report, 100 μg/kg/min of lidocaine CRI was sufficient to control intraoperative pain, probably because dogs were not undergoing laparotomy and traction on mesovarium to perform sterilization. Furthermore, we found that administration of lidocaine infusion at doses 100 μg/kg/min helped to prevent the sympathetic response to surgical stimulation without causing clinically significant hemodynamic instability, as demonstrated by the absence of changes of ECG, HR/min, SAP, DAP, MAP, capillary refill time and the peripheral pulse palpation monitored during the intraoperative period. However, when given as the sole analgesic, intraoperative 100 μg/kg/min lidocaine CRI was not as effective as its association with tumescent anesthesia for early postoperative pain control.

TUM is a technique that was initially used to perform liposuction under local anesthesia, which consists of the injection of such large volumes of local anesthetic until to produce swelling and firmness (tumescence) of the surgical area, recently introduced in veterinary medicine for pain management [22,24,25,26]. In this study, this loco-regional technique using a ten holes cannula resulted easy and quite fast to perform. Despite the solution was less concentrated than other studies [24,25,26] allowed easier removal of the mammary tissue and better postoperative analgesia in Group TUM and LID/TUM than Group LID. One explanation for the lack of statistical difference in duration of surgery between groups was the relative lack of familiarity with the TUM technique.

The pulse-oximetry measurements and invasive systolic and diastolic arterial blood pressures showed no significant differences between groups and low intra-group variability. The intergroup difference in invasive mean arterial blood pressures could indicate, with considerable sensitivity, a slightly lower pain sensation during surgery in the Group LID/TUM compared to the other groups.

The postoperative pain scoring data of this study show a clear beneficial effect of the combined action of TUM and lidocaine CRI in dogs undergoing mastectomy.

Similar to a previous study [35], a multiparametric ICMPS-SF pain scale was used because it is validated, developed for dogs suffering acute postoperative pain and easy to use by only one blinded evaluator that was involved to assess postoperative pain [27]. In fact, simple unidimensional tools are often not standardized and have a limited number of response options, providing inadequate information [35]. During the postoperative recovery phase, all groups showed progressive lowering of the pain score. Unlike the Group LID and the Group TUM, the lack of significance of this trend for the Group LID/TUM is due to the very low level of pain present at RT0, once the dogs had fully recovered from the sedative effect of the anesthetic drugs, and persisted for the remainder of the early postoperative period (60 min).

It is important to note that the effects of fluid infiltration in the operation field in Group TUM and in Group LID/TUM, in relation to the response to touch, one of the 6 behavioral categories of ICMPS-SF. Most of the infiltrated tissue along with the mammary gland following the TUM is removed during the surgery, but a small gelatinous fraction remains in the subcutaneous space which may be responsible for postoperative analgesia, as also hypothesized by Abimussi and coauthors (25). Therefore, this may justify the low values of response to touch items achieved by the Groups TUM and LID/TUM, unlike the Group LID, although further investigation is needed.

However, in the present study, dogs receiving rescue analgesia were scored until the end of the evaluation period (1 h). This approach may have increased the differences among groups because pain scores in dogs receiving rescue analgesia may have been artificially higher than those who did not receive rescue analgesia.

Another important aspect to consider in the interpretation of the results of this study is that the combined action of lidocaine CRI and tumescent anesthesia prevented the need for rescue analgesia in all dogs during the postoperative phase. Rescue analgesia was required in all the dogs in Group LID and in over half of the dogs in Group TUM, which may have positively influenced the trend of postoperative pain in both groups.

Some reports investigated postoperative pain in dogs undergoing mastectomy [11,36,37,38,39,40,41,42], but only two veterinary studies evaluated the perioperative effect of tumescent anesthesia technique in bitches undergoing unilateral mastectomy [24,25]. The authors concluded that the use of TUM in bitches undergoing mastectomy may be easily performed and provided beneficial effects such as improvement of immediate postoperative analgesia, absence of adverse signs, and facilitation of the surgical procedures. Similarly, the results of this study showed that TUM may be an effective alternative technique during complicated surgery where it requires wide tissue resection. It is important to highlight that the animals were neutered, and this condition caused the mammary gland to be less vascularized as a result of the decrease of endocrine stimulation. Therefore, the surgery was easier and the pain induced by surgery was less intensive. Furthermore, the use of Sonicision^®^ (Medtronic, Milan, Italy), a cordless ultrasonic dissection device, allowed to performed mastectomy with minimal intraoperative blood loss and tissue damage and shorter surgery time. In fact, unlike Credie and coauthors [24], there was no significant difference between the groups with respect to duration of surgery, anesthesia time, time to endotracheal extubation, and recovery time in this study.

The homogeneity between groups regarding the surgical aspects, together with that of the population regarding age and weight and the absence of complications, allowed to obtain a reliable study as far as the limited number of patients recruited and included could allow.

Acute pulmonary oedema is a reported complication after the use of tumescent anesthesia in humans, similar to Credie et al. report. However, this complication does not seem to cause concerns in this study, which may be due to the smaller volumes injected as compared to previous studies [24].

Hypothermia that developed in all animals during anesthesia is the most common anesthetic complication in small animals [43], although there were significant differences between groups. In fact, as we found in our study, more severe hypothermia might be expected in the Group TUM and LID/TUM than Group LID because of the infiltration of a cold tumescent solution. However, the body pre-warming before induction of anesthesia by means of a combination of an electric heating pad in conjunction with a radiant heat heating lamp [44] minimizes the heat loss in this study. Furthermore, body warming was also assured for the entire duration of the surgery, allowing to maintain body temperature within acceptable levels for general anesthesia.

This research had two main limitations: (1) no measurements of isoflurane concentration were performed, as it is considered one of the main factors potentially capable of causing a dose-dependent lowering of arterial pressure [45,46]. Despite in this study, the MAC isoflurane data were not collected, the invasive blood pressure, the capillary refill time, the peripheral pulse palpation and the hypothermia were continuously monitored during anesthesia in order to evaluate early the onset of possible hypotension. Furthermore, the isoflurane vaporizer setting was adjusted to deliver sufficient concentration for surgery based on clinical signs, including the absence of palpebral reflex, absence of jaw tone, and MAP between 60 and 90 mmHg; (2) the short duration (60 min) of the postoperative pain evaluation, allowing to draw conclusions limited to the immediate postoperative period.

## 5. Conclusions

The results of the current study suggest that all three treatments provided satisfactory antinociceptive effects during the surgery but administering of LID/TUM caused greater inhibition on the sympathetic stimulating effects, thus provided better early postoperative analgesia in dogs undergoing mastectomy. However, further clinical studies with a high number of subjects are required in order to evaluate the effectiveness of analgesic effects of lidocaine CRI in combination with tumescent anesthesia, as a part of management for surgery involving removal of a wide range of tissues, such as cutaneous reconstructive surgery. Furthermore, since the observation period of postoperative pain was limited, longer evaluation may be considered to support this conclusion. In conclusion, this technique could be considered a valid alternative approach to pain management, as a non-opioid treatment option.

## Figures and Tables

**Figure 1 animals-11-01280-f001:**
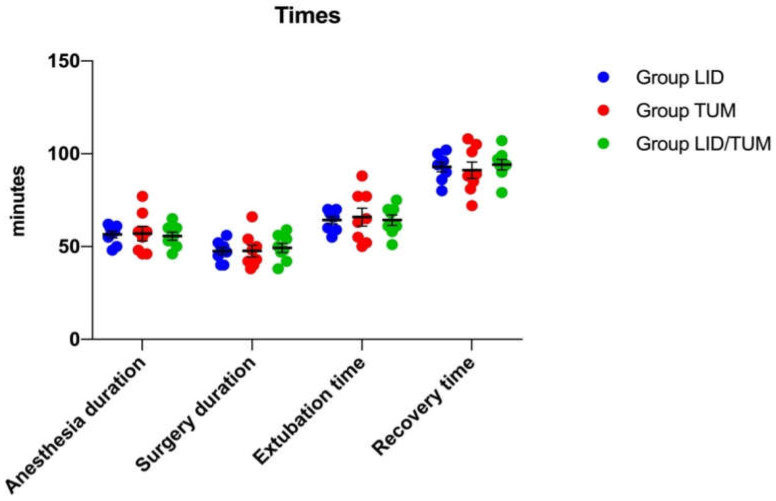
A scatter dot plot showing the anesthesia duration, the surgery duration, the time to endotracheal extubation, and the recovery time in the three groups. Blue, red and green dots indicate the single measurements scattered in groups; bars indicate means and standard errors; LID: Lidocaine group; TUM: Tumescent anesthesia group; LID/TUM: Lidocaine/Tumescent anesthesia group.

**Figure 2 animals-11-01280-f002:**
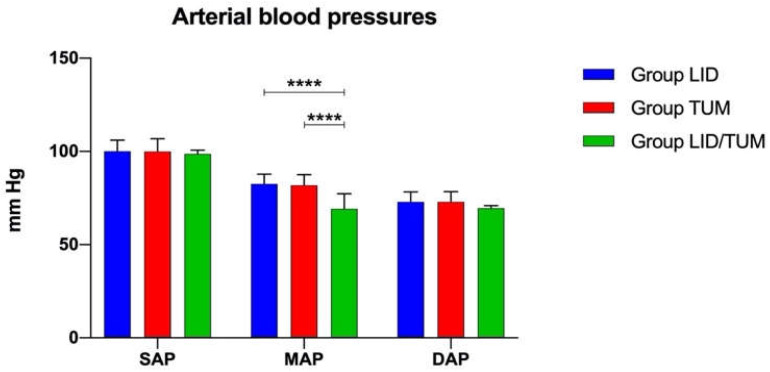
Mean values ± standard error of invasive systolic (SAP), diastolic (DAP) and mean (MAP) arterial blood pressures during anesthesia procedures in the three groups. LID: Lidocaine group; TUM: Tumescent anesthesia group; LID/TUM: Lidocaine/Tumescent anesthesia group. Asterisks indicate significant differences between groups, ****: *p* < 0.0001.

**Figure 3 animals-11-01280-f003:**
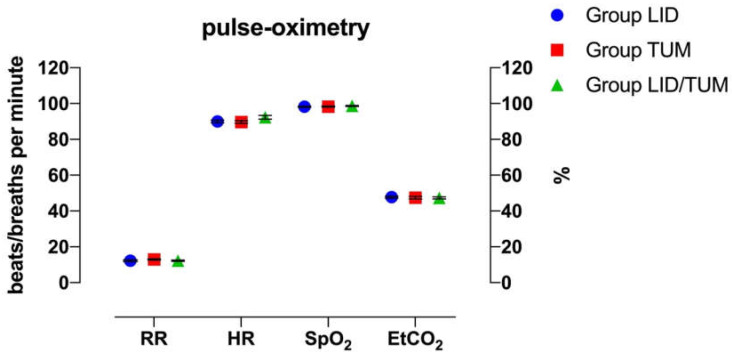
Mean values ± standard error of heart rate (HR), respiratory rate (RR), arterial oxygen saturation (SpO_2_) and end-tidal partial pressure of CO_2_ (E_t_CO_2_) during anesthesia procedures in the three groups. LID: Lidocaine group; TUM: Tumescent anesthesia group; LID/TUM: Lidocaine/Tumescent anesthesia group.

**Figure 4 animals-11-01280-f004:**
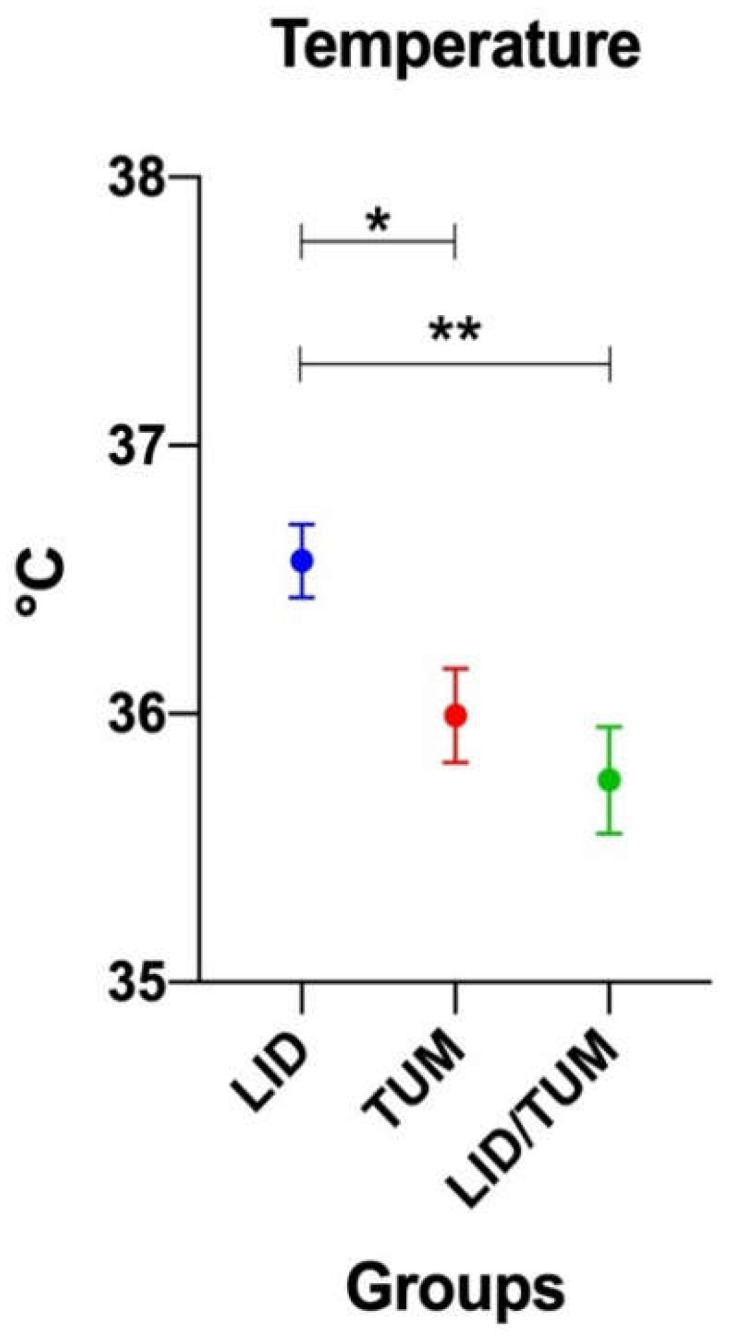
Mean values ± standard error of oesophageal body temperature during anesthesia procedures in the three groups. LID: Lidocaine group; TUM: Tumescent anesthesia group; LID/TUM: Lidocaine/Tumescent anesthesia group. Asterisks indicate significant differences between groups: *: *p* < 0.05; **: *p* < 0.01.

**Figure 5 animals-11-01280-f005:**
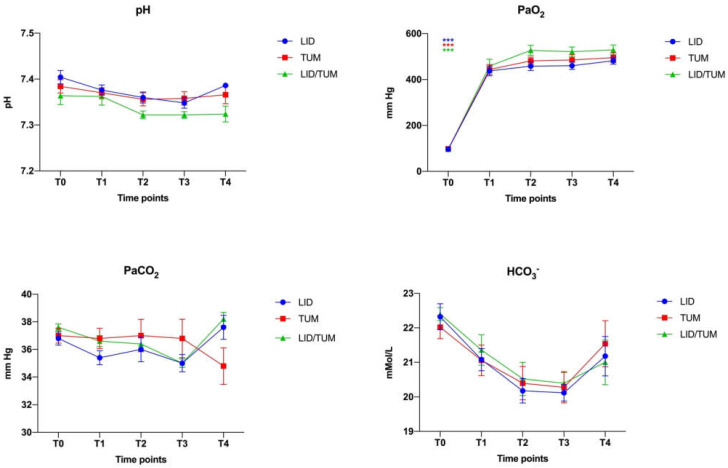
Mean values ± standard error of arterial blood pH, arterial oxygen tension (PaO_2_) arterial carbon dioxide tension (PaCO_2_) and bicarbonate concentration (HCO_3_^−^) during anesthesia in the three groups. Groups, LID: Group Lidocaine; TUM: Group Tumescent anesthesia; LID/TUM: Group Lidocaine/Tumescent anesthesia. Time points, T0: immediately after the introduction of the arterial catheter; T1: immediately after the start of the surgery; T2: 15 min after the start of surgery; T3: 30 min after the start of surgery; T4: 40 min after the start of surgery. Asteriks indicate significant differences between groups: ***: *p* < 0.001.

**Figure 6 animals-11-01280-f006:**
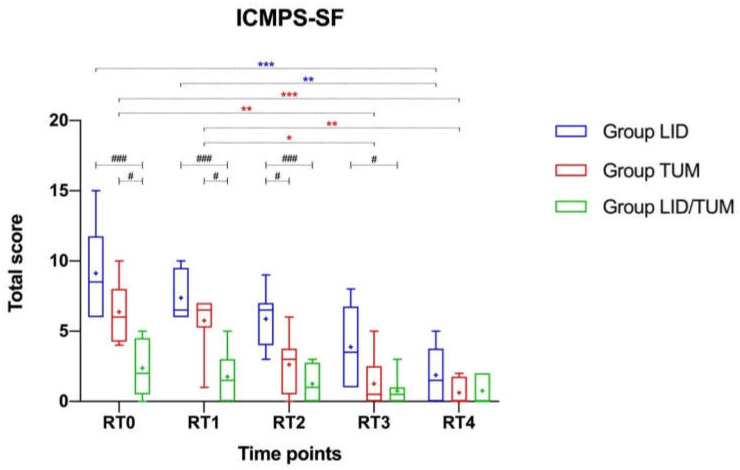
Boxplot showing the postoperative pain scores using the Italian version of the Glasgow Composite Pain Scale-Short Form (ICMPS-SF) in Lidocaine Group (Group LID), Tumescent anesthesia group (Group TUM), Lidocaine/Tumescent anesthesia group (Group LID/TUM). Time points, RT0: postoperative achievement of full recovery from the sedative effects from the anesthetic drugs; RT1: 15 min after RT0; RT2: 30 min after RT0; RT3: 45 min after RT0; RT4: 60 min after RT0. The ends of the whiskers show minimum and maximum score values; boxes show the median, the first and the third quartile; blue, red and green + show the mean values. Blue and red asterisks indicate statistical significance within each group. Black hashtags indicate statistical significance between groups. *p*-values, */#: *p* < 0.05; **/###: *p* < 0.01; ***: *p* < 0.001.

**Table 1 animals-11-01280-t001:** Mean values and standard deviations of age and weight in the three groups. LID: Lidocaine group; TUM: Tumescent anesthesia group; LID/TUM: Lidocaine/Tumescent anesthesia group.

Title 1	LID	TUM	LID/TUM	Statistics
Age	9.125 ± 2.532	9.00 ± 2.268	9.375 ± 2.200	F = 0.05338; *p* = 0.9482
Weight	14.00 ± 6.094	15.38 ± 4.897	16.50 ± 3.742	F = 0.5008; *p* = 0.6131

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
