# Peer review of "Constant Rate Infusion of Lidocaine, Tumescent Anesthesia and Their Combination in Dogs Undergoing Unilateral Mastectomy"

_animals, 2021, doi:10.3390/ani11051280_

Round 1

Reviewer 1 Report

This manuscript provide useful information for the veterinarians who perform this type of surgery. It also provide alternative methods for pain management for dogs undergo mastectomy. However, the manuscript needs some editorial revision. Please see attached file for changes recommended by this reviewer.  

Author Response

Dear Reviewer,

we thank very much for your constructive comments. as suggested by you, we tried to improve the quality of the manuscript on these bases.

As requested, all changes made to the manuscript, in comparison to the previous version, have been reported in the text with “track changes” function in microsoft word.

Below, the answers of the authors to the comments are written in blu letters.

line 15: change cri of lidocaine to lidocaine cri
line 17: change the authors found that the association of the two technique to the authors concluded that the combination of the two techniques
line 18: change lidocaine by a cri to lidocaine cri

lines 21-22: change that consists in to which consists of
line 22: change injection of such large volumes of local anesthetic until to produce swelling and to injection of large volumes of local anesthetic to produce swelling and

lines 24-25: change lidocaine administered by a cri, of either tum, or their combination, to lidocaine cri, of either tum, or their combination (lid/tum)

line 28: change much more inhibit sympathetic stimulating providing better perioperative analgesia to induced greater inhibition on sympathetic stimulating effect, thus, providing better perioperative analgesia

line 40: change complex surgeries, as mastectomy, require multimodal pre-emptive analgesia to especially after a major surgery such as mastectomy, requires multimodal requires multimodal pre- emptive analgesia

line 41: change both systematically and locally or regionally administered drugs to both systematical and local or regional administration of analgesics

lines 44-45: change the in-travenous use to the in-travenous (iv) use
line 46: change for use in the majority of surgical procedures to has been used for painful surgical procedures

line 49: change subcutaneous tissue to subcutaneous tissues; change dilute to diluted
line 51: change local solution to local anesthetic solution
line 52: change this practice of injecting was described using the klein’s cannula to this technique of injection uses klein’s cannula
line 53: change in cats submitted to in cats underwent
line 54: change demonstrating the capacity to facilitate the surgery procedure to demonstrating the effectiveness of the technique to facilitate the surgery procedure
line 55: change the addition of tum by cri of lidocaine to the addition of tum to lidocaine cri

line 64: change owner of all the dogs to owner of the dogs included in this study.
line 65: change and weighing between seven and twenty-two kilograms presenting to with body wights between seven and twenty-two kilograms presented
line 66-67: change undergoing unilateral mastec-tomy for mammary tumors to for unilateral mastec- tomy due to mammary tumors
line 68: change pre-operative to preoperative
lines 71-72: change to assess their health status which was classified as iii or less 71 according to the american society of anesthesiologists (asa) to to assess their health status. dogs with health status classified as american society of anesthesiologists (asa) iii or less were included in this study.
line 73: change dogs not spaying or obese to dogs not spayed or were obese

lines 75-76: change infiltrating large (over 5 cm) masses, and asa more than iii to infiltrated large masses (over 5 cm), and a health status greater than asa iii

line 77: change placed into a cephalic vein to placed in a cephalic vein
line 81: change electric heating pad to electrical heating pad
line 82: change a second iv catheter was applied to a second iv catheter was placed
line 84: change induction of anesthesia was performed to induction of anesthesia was produced

line 89: change 10 ml kg hour to 10 ml/kg/hour

line 96: change cri of 100 mg/kg/min associated to tum to lidocaine cri of 100 mg/kg/min in addition to tum

All suggested changes revisited in the text.

lines 97-98: change group lid/tum (n=8): a loading dose of lidocaine (2 mg/kg) followed by a cri of 100 mg/kg/min associated to tum to group lid/tum (n=8): a loading dose of lidocaine (2 mg/kg followed by a cri of 100 mg/kg/min ) in addition to tum
question:

  1. does the cri diluted lidocaine solution for tum not administered bolus but over a period of time ?
  2. for the three treatment groups:
    1. group lid: loading dose lidocaine then lidocaine cri
    2. group tum: loading dose lactated ringer’s solution then lidocaine cri +tum
    3. group lid + tum: loading dose lidocaine, lidocaine cri + tum

are these descriptions of the treatment groups correct? if so, then it needs to make

clear in the materials & methods.

We improved the group description in the text.

line 106: change spo2 to spo2
line 107: change (t c°) to (t °c); change co2 to co2; change etco2 to etco2
line 108: change pao2 to pao2
line 109: change paco2 to paco2; change hco3- to hco3-
lines 111-112: change using (i-stat system, abbott) to using i-stat system (abbott)
line 113: change the local anesthetic solution was prepared to mix 40 ml of 2% lidocaine plus 20 mg/ml adrenaline (lidocaine 2%, ati, italy) in a refrigerated (8°c) lactated ringer’s solu-tion to the local anesthetic solution for tum was prepared by mixing 40 ml of 2% lidocaine plus 20 mg/ml adrenaline (lidocaine 2%, ati, italy) into a refrigerated (8°c) lactated ringer’s solu-tion

line 115: change the local anesthetic solution to the final local anesthetic solution
line 120: change under the skin within two incisions created cranial and caudal to under the skin with two incisions created cranial and caudal
line 125 change post-operative to postoperative
line 129: change mobility to mobility
lines 131-132: change (t0), and following 15 131 (t1), 30 (t2), 45 (t3), and 60 minutes (t4) to (rt0), and following 15 (rt1), 30 (rt2), 45 (rt3), and 60 minutes (rt4)
line 141: change used to comparing to used to compare
line 148: change and weighing to with body weights

lines 165-166: change among groups; instead, the map to between groups. on the contrary, the map; change lid/tum group to group lid/tum

lines 166-167: change both the lid group 166 (82.56±5.305 mmhg; p<0.0001) and the tum group (81.87±5.643 mmhg; p<0.0001) to both the group lid (82.56±5.305 mmhg; p<0.0001) and the group tum (81.87±5.643 mmhg; p<0.0001)
line 169-170: change no significant differences between groups were showed for hr, rr, spo2 and etco2 (figure 3) to there was no significant differences observed for hr, rr, spo2 and etco2 between groups (figure 3)

line 177: delete no significant differences between groups were showed for hr, rr, spo2 and etco2 (figure 3), repeat line 169-170 sentence
line 184: change among to between
line 185: change lid group to group lid

line 186: change groups tum and lid/tum to group tum and group lid/tum
line 193-195: change arterial blood ph, pao2, paco2 and hco3- were not significantly different among groups. besides, they did not show significant differences comparing the study time-points within each group (figure 5). to arterial blood ph, pao2, paco2 and hco3- were not significantly different between groups. in addition, there was no significant difference comparing the study time-points within each group (figure 5).

line 205: change post-operative to postoperative
lines 206-238: change all t0, t1, t2, t3, t4 to rt0, rt1, rt2, rt3, rt4.
lines 214-241: change all lid group, tum group and lid/tum group to group lid, group tum and group lid/tum

line 224: change decrease in pain score to decrease in pain score at different time points

line 245: change surgical patient assistance to postoperative management for surgical patients
line 246: change failure to recognize to failure to recognize pain
line 247: change change in nervous system’s to changes in nervous system’s

line 249: change communicate, the complexity of pain perception and variation in behavioral reactions to communicate and the complexity of pain perception and variation in behavioral reactions
line 252: change post-operative to postoperative
line 253: change as respiratory depression, to such as respiratory depression,
line 254: change urinary retention, constipation to urinary retention, and constipation
line 256: change carry out to produces
line 257: change by the block of sodium channels in neural tissues, interrupting neuronal transmission to by blocking the sodium channels in neural tissues, thus, interrupting neuronal transmission
lines 258-259: change lidocaine’s in-travenous administration to iv administration of lidocaine
lines 259-260: change improve acute postoperative pain management to reduce acute postoperative pain
lines 260-261: change intraoperative cri of lidocaine to intraoperative lidocaine cri

line 261-262: change in this report, cri of 100 mg/kg/min lidocaine to in this report, 100 261 mg/kg/min of lidocaine cri
line 262: change intra-operative to intraoperative

All suggested changes revisited in the text.

lines 265-266: explain how lidocaine infusion prevent the sympathetic response to surgical stimulation without causing clinically significant hemodynamic instability

We add in the text the following paragraph: “as demonstrated by the absence of modification of ecg, hr/min, sap, dap, map, capillary refill time and the peripheral pulse palpation monitored during intraoperative period”.

line 272: change lid/tum group to group lid/tum
lines 276-280: change all lid group, tum group and lid/tum group to group lid, group tum and group lid/tum
line 278: change very low level of pain already showed at t0 to very low level of pain presented at rt0

lines 279-280: change and maintained for the whole postoperative follow-up to and persisted for the remainder of the postoperative period
line 282: change cri of lidocaine and tumescent anaesthesia allowed to avoid rescue analgesia to lidocaine cri and tumescent anaesthesia which prevented the need for rescue analgesia

lines 284-285: change rescue analgesia was instead required in all the dogs of the lid group and in over half of 284 the tum group: this could have positively influenced the trend of postoperative pain in 285 both these groups to rescue analgesia was required in all the dogs in group lid and in over half of the dogs in group tum, which may have positively influenced the trend of postoperative pain in both groups

line 287: change investigating to investigated
lines 290-292: change the use of tum in bitches undergoing mastectomy may be easily performed, improves immediate postoperative analgesia, not produces any adverse signs and facilitates the surgical procedure to the use of tum in bitches undergoing mastectomy may be easily performed and provided beneficial effects such as improvement of immediate postoperative analgesia, absence of adverse signs, and facilitation of the surgical procedures
line 292: change in the same way, to similarly,
line 293: change complex surgery to complicated surgery

lines 295-296: change this condition allowed that the mammary gland was less vascularized following the decrease of endocrine stimulation to this condition caused the mammary gland to be less vascularized as a result of the decrease of endocrine stimulation
lines 296-297: change the pain stim-ulated by surgery to the pain induced by surgery
lines 298-299: change allowed to performed mastectomy minimizing intraoperative blood loss and tissue damage, shortening the time to perform surgery to allowed to perform mastectomy with minimal intraoperative blood loss and tissue damage and shorter surgery time
lines 301-302: change with respect duration of surgery, anesthesia time and time to endo-tracheal extubation to with respect to duration of surgery, anesthesia time and time to endo-tracheal extubation in this study

lines 307-309: change although acute pulmonary oedema is a reported complication after the use of ta in 307 humans, similarly in credie et al. report, this aspect does not seem to be important, 308 probably because smaller volumes are injected than previous studies to acute pulmonary oedema is a reported complication after the use of tumescent anesthesia in humans, similar to credie et al. report. however, this complication does not seem to cause concerns in this study, which may be due to the smaller volumes injected as compared to previous studies.
lines 315-316: change by mean a composite method comprising an electric heat pad in con- 315 junction with radiant heat heating lamp [45] allowed the limitation in heat loss to by means of a combination of an electrical heating pad in con-junction with radiant heat heating lamp [45] minimize the heat loss in this study
line 320: change all the three treatments to all three treatments
lines 321-323: change administering of lid/tum much more inhibit sympathetic stimulating providing better perioperative analgesia in dogs undergoing mastectomy to administering of lid/tum caused greater inhibition on the sympathetic stimulating effects, thus provided better perioperative analgesia in dogs undergoing mastectomy
lines 323-326: change in conclusion, this technique can be considered a valid alterna-tive approach to pain management, encouraging further clinical studies to evaluate the analgesic effects of lid in cri in combination with tum, as an aid in surgery with wide removal of tissue, as cutaneous reconstructive surgery to in conclusion, this technique can be considered a valid alterna-tive approach to pain management. however, further clinical studies are required to evaluate the effectiveness of analgesic

effects of lidocaine cri in combination with tumescent anesthesia, as part of the pain management for surgery involving removal of wide range of tissues, such as cutaneous reconstructive surgery

All suggested revisited as request in the text.

Reviewer 2 Report

This is an interesting study reporting the pain relieving efficacy of the use of Lidocaine CRI in combination to local infiltration of lidocaine for mastectomy in mammary tumor affected bitches.

Main limitations: 1) for intraop assessment, isoflurane conc. should have been monitored to ensure a certain degree of anaesthetic stability/cardiovascular consequences of variable iso conc are massive 2) Postop pain evaluation limited to 60 min after recovery, which strongly limit the general applicability/reliability of the results.

Detailed comments follow.

Abstract

No results are reported. It has to be clearly stated that only 1 hour postop has been evaluated.

Intraop data strongly depend from the anaesthetic conc used,

Introduction

Line 43: the sentence about the role of lido (given IV) in the control of the sympathetic response should be together with the one explaining why lido CRI is beneficial in general anaesthesia, not in the first sentence introducing lido in general.

Line 51: what does it means mechanical technique? Also injection is a mechanical technique…Provide a short explanation of the method and avoid the use of the word mechanical

Line 55-59: check English spelling of whole paragraph.

Furthermore, important to clearly declare the aims. Intraoperative stability and early postop pain relief/need of rescue analgesia should be mentioned. The wording "influenced cardiopulmonary function" is not clear if meant in respect to intraop pain.

Material and methods

A general point: the manuscript is submitted in February 2021…why is the approval dating 2021? In general, the ethical approval has to be collected before study begin. You do not mention when the cases were collected, this should be declared in the manuscript. Also, sample size calculation is not mentioned. Why did you selected 8 animals per group? Sample size calculation and study power should be mentioned. 

Line 73: not spayed?

Line 84: provide a propofol dose range if the dose was not fixed

Line 87: expired isoflurane concentration was not monitored, right? Isoflurane has a strong effect on blood pressure, essential to keep it constant/under control in order to use blood pressure as measure of introp pain. 

Line 93: provide details about randomization method

Lines 94-99: unclear what was administered and how. For each drug/volume/dose, specify way of administration (IV or local infusion), also provide details about the administration of placebo (ringer?)

The potential effect of fluid infiltration in the operation field has to be taken into account in the discussion. Does Ringer locally infiltrated increase sensitivity of the area in the early postop? It might be…

Where does the dose of 100 mcg/kg/min lidocaine comes from? Provide a reference. Usually administered introp doses are lower…

Lines 113-123: I would place this paragraph (which need quite extensive English revision) about preparation of infiltration solution etc before the monitoring paragraph, immediately following the groups assignment paragraph. The infiltration of placebo is not described?

What do you mean with equivalent technique?

Line 125: assessed should be substituted with attributed

Line 130: was postop pain evaluated only for 1 hour? This time is really short, postop pain is usually evaluated at least 24 h post op. This short time interval has to be clearly mentioned as it strongly affect the meaning of the obtained results. It cannot be stated that the treatment reduce postop pain in general, "up to 1 hour postop" has to be mentioned everywhere, including abstract. Otherwise misleading conclusions could be drawn.

Results

Line 148: this is already said in mat and met. It would be important to state how many dogs were considered for inclusion and how many were finally included. Also, the period in which this cases collection took place has to be mentioned (over 3 years, over 2 months??).

Introperative pain assessment: blood pressure can strongly be affected by iso concentration. How did you monitor anaesthetic depth? Any attempt to control for iso conc? This is very important for the credibility of the intraop results. Why only MAP affected and not SAP and DAP? How do yu explain this discrepancy? Can you adress this in the discussion? Is the difference clinically significant? Can you really talk about better introap analgesia based on your findings?

Can you say something about the response to palpation? I know that this is one of the items evaluated by the score, but it could be interesting to evaluate this item separately, as we don't know if the infusion of placebo in the surgery area might have affected the final results. Was the sensitivity of the region one of the main affected parameters? Or something else?

Line 168: text to be deleted ??

Line 235-241: This paragraph is unclear and should be completely rewritten. Is this rescue analgesia administered within the 60 min of assessment? Which is the time frame considered?

Discussion

You used only the Glasgow pain score for postop pain evaluation. This is a validated tool, but can you discuss it shportly in relation to your surgery/population? Why this tool and not another? Advantages/disadvantages? Using only 1 tool could be seen as study limitation.

How can you justify that you looked only at this brief time period after surgery? What happened to the dogs later on? When did they go home? Observation time interval has to be mentioned as limitation of the study.

Line 261-263: this is a strange argument, in the introduction you talked about the intense pain produced by mastectomy, here it seems to go in the opposite direction. There is no reference for the IV Lido dose chosen.

Line 267 and 270-272: "introp pain " I would specify that you evaluated HR and BP stability as surrogate for introp pain evaluation. Unfortunately, as already stated, iso conc might play a big role here.

Lines 273-274: again you need to specify that you looked at a very limited time frame postop

General conclusion about postop pain reduction might be different if looking at longer time frame. Effect of local ringer infusion?

Line 296-297: part of this information has to go in the method. "surgery was easier and pain less intensive…" . Here a comparison term is missing. Compared to what? Are your results in contraposition to results from others?

Line 308-309: this sentence is not clearly formulated. Why not important? You should rather say that you did not observe this complication.

 Conclusion

I would suggest to reformulated the conclusion according to your results (here you say that intraop was the same for all groups while you previously stated that it was better in the combined group…so try to be more consistent and adhere to your findings (including the "early postop period" mentioning).

Author Response

Reviewer 2

Dear Reviewer,

we thank very much for your constructive and exhaustive comments. As suggested by you, we tried to improve the quality of the manuscript on these bases.

As requested, all changes made to the manuscript, in comparison to the previous version, have been reported in the text with “track changes” function in Microsoft Word.

Below, the answers of the authors to the comments are written in blue letters.

This is an interesting study reporting the pain relieving efficacy of the use of Lidocaine CRI in combination to local infiltration of lidocaine for mastectomy in mammary tumor affected bitches.

Main limitations:

1) for intraop assessment, isoflurane conc. should have been monitored to ensure a certain degree of anaesthetic stability/cardiovascular consequences of variable iso conc are massive

See next comment

 2) Postop pain evaluation limited to 60 min after recovery, which strongly limit the general applicability/reliability of the results.

As you suggested, we clarified in the text that we referred to early postoperative pain

Detailed comments follow.

Abstract

No results are reported. It has to be clearly stated that only 1 hour postop has been evaluated

We improved the abstract, following your suggestions.

Intraop data strongly depend from the anaesthetic conc used. 

In this study, the MACiso was not constantly measured and the results of this data have been omitted because reputed partially incomplete, despite we are aware of its importance. We commented this limitation in the text.

Introduction

Line 43: the sentence about the role of lido (given IV) in the control of the sympathetic response should be together with the one explaining why lido CRI is beneficial in general anaesthesia, not in the first sentence introducing lido in general.

This point revisited as request.

Line 51: what does it means mechanical technique? Also injection is a mechanical technique…Provide a short explanation of the method and avoid the use of the word mechanical

This point revisited as request

Line 55-59: check English spelling of whole paragraph.

We corrected English, also in accordance with Ref.1

Furthermore, important to clearly declare the aims. Intraoperative stability and early postop pain relief/need of rescue analgesia should be mentioned. The wording "influenced cardiopulmonary function" is not clear if meant in respect to intraop pain.

This point revisited as request.

Material and methods

A general point: the manuscript is submitted in February 2021…why is the approval dating 2021? In general, the ethical approval has to be collected before study begin. You do not mention when the cases were collected, this should be declared in the manuscript.

The manuscript was submitted 2 March 2021 and the ethical approval was obtain in 9 February 2021, not before study begin. In December we start to collect the cases to perform the study in order to start as soon as possible.

Also, sample size calculation is not mentioned. Why did you selected 8 animals per group? Sample size calculation and study power should be mentioned. 

Ok, we added these information

Line 73: not spayed? Yes, it was corrected.

Line 84: provide a propofol dose range if the dose was not fixed

We added this information

Line 87: expired isoflurane concentration was not monitored, right? No, it wasn’t, but we tried to justify this limitation.

 Isoflurane has a strong effect on blood pressure, essential to keep it constant/under control in order to use blood pressure as measure of introp pain. 

We are absolutely agree with your assertion and we know the role of isoflurane on blood pressure and the role of intraoperative analgesia on the isoflurane MAC-sparing effects. We would try to process the MACiso data collected.  However, hypotension (invasive pressure) or hypothermia (esophageal probe) did not occur and the capillary refill time and peripheral pulse palpation was monitored during anesthesia to evaluate early the onset of a possible hypotension

Line 93: provide details about randomization method

We added this information

Lines 94-99: unclear what was administered and how. For each drug/volume/dose, specify way of administration (IV or local infusion), also provide details about the administration of placebo (ringer?)

We tried to better clarify in the text

 The potential effect of fluid infiltration in the operation field has to be taken into account in the discussion. Does Ringer locally infiltrated increase sensitivity of the area in the early postop? It might be…

This point was better clarify in the text.

 Where does the dose of 100 mcg/kg/min lidocaine comes from? Provide a reference. Usually administered introp doses are lower…

We found that intraoperative lidocaine CRI dose is between 50 and 200 mg/kg/min. We added in the text some references to justify this dose.

Lines 113-123: I would place this paragraph (which need quite extensive English revision) about preparation of infiltration solution etc before the monitoring paragraph, immediately following the groups assignment paragraph.

It has been done.

The infiltration of placebo is not described?

It has been done

What do you mean with equivalent technique?

It was an error copied from a file that was still not corrected.

Line 125: assessed should be substituted with attributed

ok, done.

Line 130: was postop pain evaluated only for 1 hour? This time is really short, postop pain is usually evaluated at least 24 h post op. This short time interval has to be clearly mentioned as it strongly affect the meaning of the obtained results. It cannot be stated that the treatment reduce postop pain in general, "up to 1 hour postop" has to be mentioned everywhere, including abstract. Otherwise misleading conclusions could be drawn.

We are absolutely agree with your assertion. The changes have been done.

Results

Line 148: this is already said in mat and met. It would be important to state how many dogs were considered for inclusion and how many were finally included. Also, the period in which this cases collection took place has to be mentioned (over 3 years, over 2 months??).

Ok, done.

Introperative pain assessment: blood pressure can strongly be affected by iso concentration. How did you monitor anaesthetic depth? Any attempt to control for iso conc? This is very important for the credibility of the intraop results.

We answered above and added in the text this limitation of the study.

 Why only MAP affected and not SAP and DAP? How do you explain this discrepancy? Can you address this in the discussion? Is the difference clinically significant? Can you really talk about better introap analgesia based on your findings?

We considered the difference no clinically significant.

Can you say something about the response to palpation? I know that this is one of the items evaluated by the score, but it could be interesting to evaluate this item separately, as we don't know if the infusion of placebo in the surgery area might have affected the final results. Was the sensitivity of the region one of the main affected parameters? Or something else?

We added this information in the discussion.

Line 168: text to be deleted ?? yes, we are so sorry. The line is 268…..

Line 235-241: This paragraph is unclear and should be completely rewritten. Is this rescue analgesia administered within the 60 min of assessment? Which is the time frame considered? ok, done

Discussion

You used only the Glasgow pain score for postop pain evaluation. This is a validated tool, but can you discuss it shortly in relation to your surgery/population? Why this tool and not another? Advantages/disadvantages? Using only 1 tool could be seen as study limitation.

This point revisited as suggest.

How can you justify that you looked only at this brief time period after surgery? What happened to the dogs later on? When did they go home? Observation time interval has to be mentioned as limitation of the study.

This point explained in the revisited text: However, in the present study, dogs receiving rescue analgesia were scored until the end of the evaluation period (1 hours). This approach may have increased differences among groups because pain scores in rescued dogs may have been artificially higher. Moreover, there wouldn't be dogs that after the first hour would have needed rescue analgesia, and this aspect could allow the possibility to send the animals home soon.

 Line 261-263: this is a strange argument, in the introduction you talked about the intense pain produced by mastectomy, here it seems to go in the opposite direction.

We modified the paragraph according to your suggestions.

 There is no reference for the IV Lido dose chosen.

The references are 15 and 37 (in the references paragraph of the modified text).

 Line 267 and 270-272: "introp pain " I would specify that you evaluated HR and BP stability as surrogate for introp pain evaluation. Unfortunately, as already stated, iso conc might play a big role here.

We added the following paragraph: Isoflurane concentration is the most important parameter that caused a dose-dependent decrease of arterial pressure. Despite in this study the MAC isoflurane data are not collected, the invasive blood pressure, the capillary refill time, the peripheral pulse palpation and the hypothermia were continuously monitoring during all period of anesthesia to evaluate early the onset of a possible hypotension. Furthermore, the isoflurane vaporizer setting was adjusted to deliver sufficient concentration for surgery based on clinical signs, including absence of palpebral reflex, absence of jaw tone, and MAP between 60 and 90 mmHg. (with references….).

Lines 273-274: again you need to specify that you looked at a very limited time frame postop

We specified this limited period of observation.

General conclusion about postop pain reduction might be different if looking at longer time frame. Effect of local ringer infusion?

No side effects occurred in any of the animals. We reported this data in the discussions. 

Line 296-297: part of this information has to go in the method. "surgery was easier and pain less intensive…" . Here a comparison term is missing. Compared to what? Are your results in contraposition to results from others?

No, the results are superimposable to previous study.

 Line 308-309: this sentence is not clearly formulated. Why not important? You should rather say that you did not observe this complication.

We tried to formulate again this sentence.

 Conclusion

I would suggest to reformulated the conclusion according to your results (here you say that intraop was the same for all groups while you previously stated that it was better in the combined group…so try to be more consistent and adhere to your findings (including the "early postop period" mentioning).

As you suggested, we reformulated the conclusion.

Round 2

Reviewer 1 Report

Minor changes:

Line 40: change grater to greater

Line 71: change underwent to unilateral to underwent

Line 78: change by lidocaine by a CRI to lidocaine CRI

Lines 102-103: change greater ASA III to greater than ASA III

Line 139: change 1,65 mm to 1.65 mm

Line 326: change another 4 hours infusion to another 4 hours of infusion

Lines 333-334: change absence of modification of to absence of changes in Line 368: change It is significant to highlight the effect to It is important to note that the effect 

Line 372: change and may be responsible to which may be responsible

Line 375 change group LID to Group LID

Lines 377-378: change This approach may have increased differences among groups because pain scores in rescued dogs may have been artificially higher to This approach may have increased the differences among groups because pain scores in dogs receiving rescue analgesia may have been artificially higher than those did not receiving rescue analgesia.

Line 379-380: delete sentence "Moreover, there wouldn't be dogs ......to send home soon."

Line 384: of Group LID to in Group LID

Author Response

Dear Reviewer,

we thank very much for your constructive comments. as suggested by you, we tried to improve the quality of the manuscript on these bases.

as requested, all changes made to the manuscript, in comparison to the previous version, have been reported in the text with “track changes” function in microsoft word.

Line 15 - include "regional" or "local" in the description of TUM
Line 27 in abstract - TUM needs to be defined in its first usage in the abstract. delete "either" before TUM
please include that the dogs were maintained with isoflurane anesthesia
Line 39 - administration instead of administering
line 61 - insert - "and" before has been used
line 71 -  "in cats underwent to unilateral" should read: in cats that underwent unilateral
line 73 - replace surgery with surgical
Line 89 - although perhaps obvious - it should say female dogs
Line 99 Exclusion criteria for this study were dogs not  spayed or were obese and with abnormal laboratory data, arterial hypertension or congestive heart failure, renal or hepatic dysfunction, pulmonary metastases, inflamed or ulcerated tumors, infiltrating infiltrated large masses (over 5 cm) masses, and a healthy status greater ASA III.
should read
Dogs were excluded from the study if they were not spayed, were obese or had abnormal laboratory data, arterial hypertension, congestive heart failure, renal or hepatic dysfunction, pulmonary metastases, inflamed or ulcerated tumors, infiltrating infiltrated large masses (over 5 cm) masses, or an ASA health status of greater than III.
line 174 what were the criteria to determine the dogs had fully recovered from the sedative effects?

WE ADDED: ONCE THE DOGS HAD FULLY RECOVERED CONSCIOUSNESS AND WERE ABLE OF STANDING
line 194 - arm not arms
line 214 were not was
line 259 - it is either significant or not - not slightly significant - so please remove "slightly"
line 325 - reported, not referred
Line 344 study [25,26], allowed anyway an easier - should read: studies [25, 26] allowed easier
line 346 . However, there were not statistical differences in the duration of surgery between groups, probably because there was still not enough familiarity with the technique.
consider: One explanation for the lack of statistical difference in duration of surgery between groups was the relative lack of familiarity with the _TUM?_______ technique
line 353 - any alternative explanations for the difference in MAP?

WE DIDN’T FIND OTHER EXPLANATIONS FOR THIS DATA.
line 357 - In this study, a multiparametric ICMPS-SF pain scale was used, in accordance with 357
similar study [38], because it is validated . . .  Consider Similar to a previous study [38], a multiparametric ICMPS-SF pain scale was used because it is validated .. .
line 361 -  limited number of response option - should be options
line 364 present not presented
line 377 hour not hours
line 398 - result not results
